# Economic Development, Informal Land-Use Practices and Institutional Change in Dongguan, China

**Yingmin Huang [1], Desheng Xue [2] and Gengzhi Huang [2,*]**

[1] School of Architecture and Design, Jiangxi University of Science and Technology, Ganzhou 341000, China; huangyingmin693@163.com
[2] School of Geography and Planning, Sun Yat-Sen University, Guangzhou 510275, China; eesxds@mail.sysu.edu.cn
[*] Correspondence: hgzhi@foxmail.com

**Abstract:** This paper is engaged with the critical perspective that highlights the role of the state in the production of urban informality by examining the dynamics of informal land-use practices in Dongguan, China since 1978. Based on in-depth interviews and archival analysis, the relationship between informal land development, the state, and land institution change has been revealed. Our findings show that informal land development is practiced by village collectives from below in Dongguan as a response to the absence and limitation of the national land law. The local government handles the informality in a pragmatic way that serves the goal of economic development in different historical conditions by actions of encouraging, tolerating, and authorizing, suggesting that the definition of informality is not a neutral classification. It is argued that while informality represents people's creativity in dealing with practical problems, when and to what extent it can be tolerated, formalized, and absorbed depends on the intention of the state in a specific historical context.

**Keywords:** informal land-use practice; institutional innovation; urban informality; state governance; the PRD

## 1. Introduction

Urbanization in China has progressed rapidly since the reform and opening up in 1978, with the proportion of the urban population growing by 60.60% and the urban built-up area increasing by 58,455.66 km² in 2019 [1]. A large amount of land resources was developed for the sake of industrialization and urban construction, resulting in a dramatic change of land-use structure over the country [2]. At the local level, the city government fanatically developed land as a tool for promoting urban expansion and economic growth and as a source of local financial revenue [3]. This land-led development constitutes a key facet of China's urban and economic growth in the post-reform era [4,5]. However, rarely known is the informal land-use practice behind the dramatic urban transformation, which was pervasive in China and mostly occurred in urban fringe areas at the township level [6]. A high proportion of rural land conversion for industrial and housing uses has occurred through informal channels [7]. According to the statistics of the Ministry of Land and Resources of China (MLRC) in 2014, there were 56,926 cases of unapproved land use, involving 348.82 km² of land area [8]. In Guangdong Province, which was a demonstration zone for institutional experimentation in China [9], there were 7129 cases of unapproved land use, involving 20.55 km2 of land area, including 2.92 km2 of arable land in 2014. These data suggest that China's rapid urbanization cannot be fully understood without considering informality in land development.

Informal land-use practices in China have drawn increasing attention from both the public and academics [10,11]. This term generally refers to urban development on land

without land-use permission or planning approval from the state or development that does not comply with land-use planning and development regulations [12]. Informal land use can be understood as the spontaneous response of society to the absence or imperfection of land institutions. It is a pragmatic practice from below, involving the circumvention of and incomplete complying with formal land institutions based on practical needs [13], such as the behavior "hitting an edge ball".

Academic interest in informal land-use practices in China has mainly focused on the formal-informal dualism perspective [14,15]. Studies also examine the actors of urban development and economic growth, such as local villages, village collectives, and small enterprises, and how their interests and survival needs lead them to break land laws [12,16,17]. Little research has been concerned with the role of the state in the dynamics of informal land-use practices and their effect on the formal land system [2,18]. Following the literature of urban informality [19,20], this paper investigates forms and dynamics of informal land-use practices on the part of the local state in the Chinese context with a case study of Changan Town in Dongguan City in the Pearl River Delta (PRD). It examines why and how informal land development is practiced by the local state at the township level in different historical circumstances since 1978 and how it brings about changes to the formal land institution. The paper furthers the conceptualization of informality as forms of governance by investigating how the government navigates the relationship between informal land development and formal land system to achieve the goal of economic development in different institutional contexts. From this investigation, we detect the possibility of informality as the seed of institutional innovation. Our main argument is that informality is both a violation and a seed-of-change of formal institutions.

This research contributes to the existing literature by examining how the relationship between the Chinese state and informal land practices has evolved since the reforms of 1978, and with more recent decentralization [2,20]. Urban development and informal land practices in China have occurred in the context of political and economic transition, characterized by numerous simultaneous processes, including decentralization, marketization, and globalization, which have significantly changed the relationship between the state and city governments [21]. Moreover, because the state is an actor, this study has incorporated related literature regarding state development to understand the regulation of informal land practices in the Chinese milieu. This approach has permitted a critical understanding of informal land practices at the grassroots level by considering informality as a device that reveals the nature of the state [22].

The rest of this paper proceeds as follows. The second section discusses existing literature on urban informality and the theory of institutional change and elaborates on the article's analytical framework. The third section introduces a case study on the relationship between the state and informal land practices in Changan Town of Dongguan City in PRD, China. The final section discusses the results and their policy implications.

## 2. Literature Review: Urban Informality and Institutional Change

### 2.1. Urban Informality: A Mode of State Governance

Urban informality is a pervasive phenomenon in the global South, which has attracted attention from economic, sociological, and urban studies since the early 1970s [23,24]. Research on urban informality has evolved over more than four decades, from investigating a single discipline or territory into a comprehensive, transnational, and comparative topic. Three types of theoretical perspectives have generally guided informality research. First, the early dualist perspective viewed informality as sets of traditional and undeveloped socio-economic activities, which are divorced from formal economic sectors in developing countries [11,23]. This view emphasized a precise distinction between formal and informal sectors. The informal sector was generally associated with marginalization and poverty but not regarded as illegal; instead, it represented the survival strategies of the grassroots amid difficult living conditions, which were outside state regulations.

Therefore, identifying how to reduce informal activities through the accelerated development of formal economies was viewed as a significant concern.

Second, the New-Marxism theoretical perspective criticized dualism, arguing against the dichotomy between the formal and informal sectors [25,26]. This perspective considered the informal and formal sectors as closely connected, with the informal economy remaining a segment of modern economic systems.

Third, the neo-liberal perspective developed in the context of the acceleration of economic globalization and the prevalence of neo-liberal policies. This theoretical perspective views informality as the grassroots' spontaneous response to the state's overregulation; the excessive regulation of economic activities by the state causes the formation of the informal economy. Informality was the real reflection of the market, rather than the consequence of unemployment [27]. The economic development gap between developing countries and developed countries persists due to the developing countries' lack of formal property rights systems. As a result, the informal economy fails to be transformed into a conventional formal market [28]. However, the three perspectives generally treat the state as a background factor and, therefore, cannot fully explain informality.

Although the study of informality began with a focus on the informal sector in the 1970s, it has been extended to include informal spaces such as informal settlements, informal housing, and informal land use. There are no closed linkages between informal sectors and informal spaces [29–32]. There is still no defined concept for informality; the sole consensus is that lack of regulation leads to the formation of informality [14]. Based on these three theoretical perspectives, some research on informality has emphasized the role of the state in recent years. The appearance of the criticism governance perspective at the beginning of the 21st century introduced a novel and more profound understanding of informality [24]. Roy and AISayyad provided a comprehensive discussion of the relationship between the state's power and informal practices, introducing the concept of urban informality. There are three primary academic contributions of the criticism governance perspective [33].

First, advancing beyond the dualism perspective, the criticism governance perspective posits that formality and informality are not a simple binary opposition of legality and illegality, regulated and unregulated, or either in or out of control [14]. Rather, informality is the process of deregulation of the state. Formality and informality are not only contradictory, they are also connected with each other and one can become the other, moving the boundary between them, becoming a continuum. Informality lies within the scope of the state, rather than outside of it, and is a deregulated system rather than an unregulated one [19]. For instance, Dicken (2005) argues that Rio's favelas, far from being marginal spaces in the city, are central to the logic of urbanism and law [34].

Second, urban informality is a mode of governance. Urban informality is a flexible strategy of the state under different political, social, and economic circumstances. Urban informality can be viewed as the space practice of the state under the interaction of all actors in urban development and economic growth, such as the central state, local states, enterprises, village collectives, and villagers, rather than a simple economic sector or geographical space [20]. The production of space within a state's territory is embedded in its sovereignty [35–37], because the state has the power to determine what is informal and what is not, and the state can determine which forms of informality will thrive or fail [22]. In the case of Turkey, political authorities reconstituted the informal-formal spatial divide to support their own land claims [38].

Third, the epistemology of urban informality has shifted from bottom-up to top-down. The state contributes significantly to urban informality. As the informal is defined as the socio-economic activities that occur outside of and separately from the formal economic system, informality is typically observed in urban "grey spaces" and "shadow cities" [39]. Informality is usually regarded as the space practice of the grassroots; it is bottom-up and can be understood as a static object of study. Moreover, formality can be understood and considered a lie, or a temporary status; it is an ambivalent and uncertain

system [22]. In other words, informality exists in the core power of the state, and it is the government that sets the conditions of the possibility of informality. In contrast to the idea that informality is caused by the lack of state regulation, Roy contends that informality is generated by the state itself [40]. The state's selective enforcement of regulation, the suspension of relevant laws, and the partial authorization of informality, indicate a "calculated" informality, or a "system of deregulation" that is, in essence, a "mode of regulation." Recent studies have revealed the failure of the state to end social practices such as informal housing and street vending [41–43].

### 2.2. Institutional Change: From Informal Institution to Formal Institution

Over the last three decades, with the development of economic globalization, the social sciences have embraced new institutionalism by recognizing the centrality of institutional frameworks, when dealing with social and economic phenomena [44–47]. Institutions are one of the primary factors of production, which contribute significantly to the economic growth and reshape the capitalist milieu; institutional change is considered the fundamental source of economic growth and urban development [48].

This study utilized institutional theory to study formality and informality. According to Douglass North, institutions "are the rules of the game in a society, or more formally, are the humanly devised constraints that shape human interaction" [48]. There are two kinds of institutions: formal and informal. Formal institutions are governed by rules codified by laws, regulations, administrative orders, and administrative statutes. Informal institutions are defined as organizations that are motivated by deeply embedded values, norms, customs, and traditions [49]. Both formal and informal institutions can enable and constrain human behavior [50]. Informal institutions may also exert considerable influence on formal institutions. In fact, throughout human history, many formal institutions were established upon the foundation of informal institutions, which modified, supplemented, or extended to become the formal institution [51]. The study of the global South has discovered that informal practices not only supplement and rectify the defects of formal institutions but also become the foundation of state reform and institutional innovation [52].

Institutional change can involve the substitution of a less effective arrangement in socio-economic activities, which is an ongoing evolution from institutional imbalance toward innovation, and ultimately, the realization of institutional equilibrium. Institutional change essentially involves the transfer and redistribution of power and interests. In general, informal institutions are transformed into formal institutions. The state and local governments, as the primary founders of institutions, typically make institutional arrangements to serve the space production within their territory, based on the characteristics of institutional implementation, such as the changes within socio-economic structures and the goals of urban development [18].

While the social sciences have given greater attention to formal institutions, the study of informal institutions is by no means a new research agenda [53]. Informal institutions are equally as important as formal ones for understanding the world. How do informal institutions emerge, spread, change, and become formalized? In recent years, some research on institutional change theory has moved beyond institutional forms and has again, interjected institutional function into the discussion [54]. The functionalist approach holds that informal institutions emerge to perform essential functions, such as providing efficient solutions to problems of information or collective action [50]. For example, Helmke and Levitsky argued that informal rules may be created when formal institutions are incomplete and cannot cover certain contingencies [49]. Similarly, Tsai found that local actors devise informal coping strategies to evade the restrictions of formal institutions [55].

However, in the scholarly debates concerning informality, the term "informal" is not linked to institutions in North's work or interactions among actors. In general, the field of

urban studies has not applied the institutional perspective to the impact of urban development institutional change theory [48]. Nevertheless, drawing on Roy's urban informality and North's institutional change theory, Altrock proposed the concept of conceded informality [19]. He analyzed the connection between the two theories, which include interactions between state and local governments, enterprises, villagers, and village collectives. Not only are formal and informal institutions constructed, but also the formal and informal urban development statuses are formed in the socio-economic system. The state is viewed as the central actor, that determines the status of institutions and urban informality.

In summary, land is not only the spatial carrier of urban development but also an important tool for attracting investment. In the context of global environmental change, rapid urbanization, and sustainable development, land use has been a great concern among Chinese academics [2,42]. The reform of land systems is the most important institutional change in China since 1978, and it has had a far-reaching influence on urban development. The PRD was a typical peri-urbanization area under the bottom-up urbanization mode [16], and there have been significant informal land practices at the grassroots level in the PRD since the market-oriented institutional changes of 1978.

This process, however, has differed significantly from the conventional understanding of informal land-use practices, as a negative consequence of state-led land expropriation [6]. This research contributes to the understanding of informality as a production of the state by investigating the regulation of informal land practices in China since 1978. By exploring the state's motivations behind regulatory practices in different historical circumstances, this present study argues that the definition of informality is not a neutral classification, but rather, one made and remade by the state to satisfy its political purposes. The state is viewed as an actor and is understood to have disclosed the relationship between the regulation of informal land practices and the political purposes of the state.

Although it has been proven that the state and the land institutions are contributing more significantly to urban development and economic growth, there exists a close relationship between informal land-use practices and land institution innovation, especially in regard to the acts of the state from the perspective of criticism governance. Three questions remain unanswered. First, how did informal land practices in the PRD emerge, spread, and persist? Second, how did the state and local governments deal with large-scale informal land-use practices in different historical circumstances? Third, how did informal land institutions become authorized by the state and come to represent land institution innovation? The case study answers these questions.

## 3. Materials and Methods

### 3.1. Study Case

Changan Town is located on the south of Dongguan City, Guangdong, China, which is in the Shenzhen-Guangzhou Economic Corridor. It is known as the "world factory zone" (see Figure 1). There are 13 villages or communities under the jurisdiction of Changan Town, which covers an area of 81.5 km². The level of socio-economic development has increased rapidly since 1978, with a population size of 674,000 and 76.03 billion yuan GDP, Changan was ranked as the seventh most important industrial town of 1000 in China in 2019. However, there are pervasive informal land-use practices in Changan for numerous reasons. First, the specific administrative structure in Dongguan has four levels: city, town, village, and group. Second, the center of economic growth is at the grassroots level, especially at the levels of the town, village, and group. Third, there is a cultural tradition of significant autonomy in the villages in Guangdong; villages and the village collectives determine their affairs independently, including decisions regarding land utilization, building plans, infrastructure development, and their execution. Finally, the rapid economic growth in Changan has been supported by the sufficient supply of rural land; villagers collectives own the property rights over rural land by law, which they

mobilized to attract industrial investment to increase the villagers collectives' income in pragmatic ways that circumvented the formal land system. For these reasons, Changan Town in Dongguan City was selected for the case study, as it can illuminate the relationship between the state and informal land-use practices and institutional innovation at the grassroots level in China.

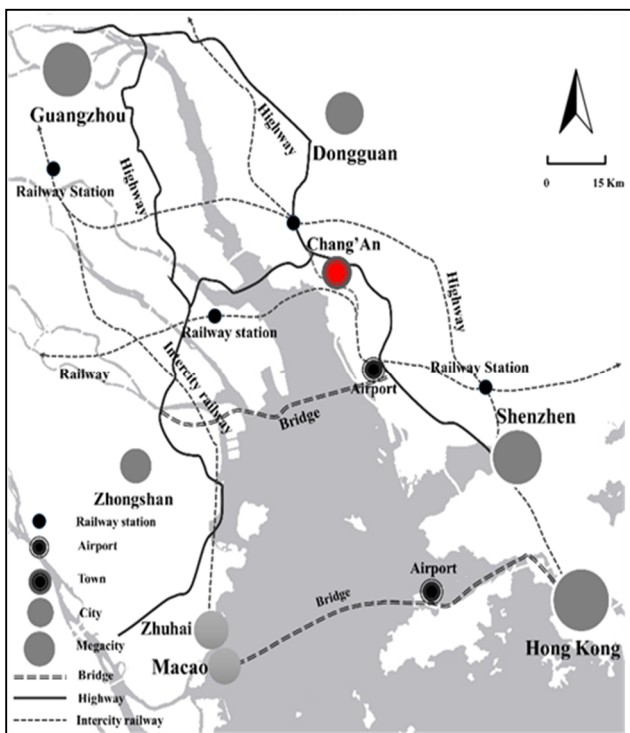

**Figure 1.** The location of Chang' An Town in the PRD.

### 3.2. Study Method

The case study focuses on three dimensions. The first dimension is the informal land, which is the core factor of production. The second dimension consists of the formal and informal land-use institutions of the state, local governments, and the grassroots level, which is the second factor of production. These two dimensions contributed significantly to the urban development and economic growth in Changan. The third dimension is the creator of the land institutions, and the primary actors in economic growth, such as the state, local governments, enterprises, villagers, and village collectives (see Table 1).

**Table 1.** The interactors of land institutional change.

| Interactor | Main Subjects |
|---|---|
| State | The central government and the Ministry of Land and Resources |
| Local state | Guangdong provincial government, Dongguan government, Changan Town government, and the planning, land, and three old transformation governments at all levels |
| Enterprises | Foreign-invested enterprises and private enterprises (like the "three-plus-one" enterprises (enterprises that process raw materials on clients' demands, assemble parts for the clients and process according to the clients' samples or engage in compensation) and the mold factories) |
| Villagers and village collectives | Villagers and village (community) collectives |

This study employed qualitative research methods, such as in-depth interviews and document analysis, to analyze the root cause and forms of informal land-use practices and their relationship with the state in Changan Town during different socio-economic and historical circumstances since 1978. First, in-depth interviews were conducted with local villagers, village cadres, and entrepreneurs, investigating how they developed informal land practices and coping strategies for managing regulations by the state and local governments. The interviews were conducted with villagers and village cadres from Xiaobian, Jinxia, and Yongtou; the managers and workers in enterprises funded by investors from Hong Kong and Taiwan were also interviewed. In addition, in-depth interviews were conducted with local officers from the urban planning and management office, the land resource management office, and the "three old (old towns, old villages, and old factories) redevelopment" office in Changan Town, investigating the roles and attitudes of the townships and city governments regarding informal land-use practices. The interviews were conducted on 10 October, 2015 and throughout February 2016. A total of 23 persons were interviewed, conversing with some individuals multiple times, culminating in more than 33 appointments, that ranged from 0.5 to 3 hours, each. The entire process was recorded with the consent of the interviewees.

Secondly, document analysis methods were employed; archives, newspapers, official statistics, historical documents, and public statements of senior officials from Beijing were analyzed in order to explore the forms of informal land-use practices and the goals and dynamic mechanisms of the changes to land institutions in different historical circumstances. The documents included *The Annals of Changan Town, Ten Preferential Treatments for Foreign Investors in Setting up Factories in Dongguan County, Measures of Guangdong Province for the Administration of the Transfer of the Right to Use the Collective Land for Construction Purposes, Some Opinions of Guangdong Province on Implementing the Three Old Transformation to Promote the Intensive Land Use, Specific Rules for the Enforcement of the Three Old Transformation in Dongguan City*, and the documentary, *Changan's Development in Past Thirty Years.* Land-use data and information were also obtained.

As discussed in the empirical section of the study, the root causes and methods of informal land-use practices during different historical circumstances were explored, including the state's motivations, macro-backgrounds, features of urban development, economic growth, and the regulatory practices concerning formal land institutions in different historical circumstances. These causes and methods can be understood by disclosing the relationship between the informal land-use practice and the political purposes of the state.

## 4. Informal Land-Use Practices, Economic Growth and State Governance in the Changan Town Since 1978

### 4.1. Deregulation and Active Support for Informal Land-Use Practices in the Early Days of the Transformation (1978–1986)

Gaps in land institutions led to the emergence of informal land-use practices in Changan in the context of deregulation, decentralization, and globalization in China. The state actively supported informal land-use practices in the early days of the reform and opening in China, with the institutional transformation from the planned economy to a market-oriented economy. The Chinese central government delegated powers to Guangdong Province, as a pilot area for the reform, to experiment with market-oriented economic institutions, to support economic growth in the PRD. Meanwhile, the government of Guangdong Province delegated powers to city, county, and town governments, to ultimately extend the grassroots level. Additionally, there was a great disparity in economic development between the PRD and Hong Kong, as Guangdong, in particular, was one of the poorest regions in China, before 1978. Under the pressure of the international industry, the institutional balance was shifting from Hong Kong to the PRD, paralleling the global development of an international division of labor. Under these circumstances, the PRD

gradually received more overseas investments from Hong Kong and Taiwan, which was initiated in the early 1980s. A consensus to promote economic development in the PRD was reached among four stakeholders: the state, local governments, investors, villagers, and village collectives.

Although China did not have market-oriented land institutions before 1978, all government levels actively supported institutional experimentation for economic growth. Growth in Changan, which was mainly driven by investments from Hong Kong, increased under a special economic model called "the three-processing and one compensation" economy. Specifically, the Hong Kong investors supplied raw materials, machinery, equipment, and product samples, while the local villagers and village collectives supplied the land and labor. The manufactured products were sold overseas, and the Hong Kong investors paid remuneration for the land and labor to the villagers and village collectives. As the villagers received more compensation through this model, they continued to supply land and labor, and additional rural collective land was converted into industrial land. This process promoted rural industrialization in Changan.

To further promote economic growth in the PRD, informal land-use practices were actively supported by states at all levels, as formal land institutions were insufficient. For example, the secretary of the Guangdong Provincial Party Committee, Ren Zhongyi, gave a public speech in 1980, actively supporting the exploration of informal practices; the speech emphasized that extant institutions should be flexible, or be replaced with new ones, if found lacking in contributions to economic growth [56]. After "the three-processing and one compensation" economic model progressed, the Guangdong Province government enacted *The Interim Provisions on Strengthening the Management of the Foreign Processing and Assembling Businesses* in 1983 to provide institutional support for foreign investment. Dongguan County government issued *Ten Preferential Treatments for Foreign Investors in Setting up Factories in Dongguan County* in 1984. Term one provided that foreign investors who established factories in Dongguan would be offered land at a preferential price, that was lower than in the Shenzhen special economic zone; investors interested in building villas in Dongguan could also benefit from the preferential land price. Moreover, the Dongguan local government developed a strategy for rural industrialization after the county was upgraded to a county-level city in 1985. All of these institutions promoted the growth of informal land-use practices because formal market-oriented land institutions in China were insufficiently appealing to investors.

Another reason for the emergence of the informal land-use practices in Changan was a persistent misunderstanding of the land institutions by the villagers and village collectives. China's constitution of 1954 stipulated that villagers and village collectives were the joint owners of the rural collective lands, which ensured their high degree of autonomy over that land. Market-oriented land institutions were lacking in the early days of the reforms, and access to industrial land was usually acquired free of charge through administrative allocation. The state enacted the Household Contract Responsibility System and subcontracted the collective land to households in 1980, which led to the mistaken belief that rural collectives were privately owned. Information gleaned from village interviews in January 2016, revealed that the villagers believed they possessed ownership rights that would allow them to change the function of the rural land. Furthermore, along with the collectives, the villagers believed there was more to gain by converting rural land to industrial land under the "three-processing and one compensation" economic model. For instance, local villagers were typically appointed as factory directors or workers, allowing them to earn greater remuneration from the land. Under these conditions, landowners typically actively supported the informal land-use practices.

The number of "three-processing and one compensation" factories continued to grow from the early days of reform in Changan, increasing to 45 by 1986. The village collectives could acquire more processing fees and remuneration from these factories than from agriculture, which they either redistributed to the villagers or used to build roads or more factories. As rural land became the key factor in attracting additional investment,

informal land-use practices, mainly changing agricultural land to industrial land, continued to increase, driven by the economic interests of villagers and village collectives. Initially, existing structures, such as ancestral halls, dining halls, and conference halls, were used for factories in Changan. Later, agricultural land was taken too, decreasing the overall area by 794.47 km² by 1985.

In an interview with the secretary of Xiaobian village in Changan in December 2015, a research interviewer learned there were two primary types of informal land-use institutions involving the grassroots and investors from Hong Kong during the early days of reform in Changan. The first type was land leases. In order to reduce the cost and lower the risk of investment, investors usually leased the village's factories when they founded their enterprises in Changan. The second type of informal land-use was land transfers. The "three-processing and one compensation" enterprises usually intended to expand their production scales after generating initial profits, but the crude factories provided by the villagers in Changan failed to meet their production needs. In such cases, the poor villagers sold the rural collective land to the Hong Kong investors, signing informal land transfer agreements, enabling the investors to design and build more advanced factories. In an effort to promote economic growth, the village collectives and the township governments typically supported these informal institutions.

### 4.2. Re-regulation and Toleration of Informal Land-Use Practices (1986–2005)

With the rapid rural industrialization in the PRD, a large amount of agricultural land was informally leased and transferred to foreign investors, then converted into industrial lands. This informal agriculture land conversion was extensive and disorderly, causing two major problems: the loss of arable land and land-based social unrest [6,7]. Although the grassroots accrued some gains from the industrialization of rural areas, they lost their farmland. The per capita farmland in China began decreasing annually, gaining the state's attention, which made the protection of farmland part of its national strategy [2]. The state issued the National Land Management Law in 1986 to regulate land use, to maintain national food security and social stability. However, informal land-use practices continued to increase in Changan after 1986 for four reasons.

First, the villagers misunderstood the nature of formal land institutions. For instance, Article 2 of the National Land Management Law stipulated that no unit or individual could legally appropriate, sell, lease, or transfer rural land. Article 39 of the law required that construction on rural land by township and village enterprises be approved by a local government above the county level and that the scale of rural construction projects should be rigorously controlled. The law was formally implemented on 1 January, 1987. In addition, to adapt to economic development, the state revised the constitution in 1988, separating land ownership and land-use rights, regulating land-use rights transactions. Although the rural land was owned by villagers and village collectives in China according to the constitution, they had no right to transfer or change the nature of rural land [12]. If the villages wanted to repurpose rural land into industrial land, the requirement stipulated the land must first be designated as urban land, owned by the state. This meant that the development rights for rural land still belonged to the state and its agents, the city governments. Although rural land use was rigorously regulated by the National Land Management Law, the villages and village collectives in Changan only had a tenuous understanding of the laws. In their opinions, they held ownership of the rural land and could transfer or use it in any way under the constitution and the Household Responsibility System.

Second, there were conflicts between the rigorous land management law and the land-use demands of villagers. Foreign direct investment (FDI) in Dongguan increased rapidly in the 1990s, fuelling rural economic growth in Changan. Given the favorable economic development opportunities, the village and village collectives endeavored to transition additional rural land into industrial land. Informal land use, especially the unauthorized conversion of cultivated lands for non-agricultural uses, is a persistent feature of

the reform era in China [7]. According to the formal land institutions, the amount of rural land that could be designated as industrial land was limited and the duration of the approval process was lengthy, it would lead to the villagers lose their land-use rights. A village cadre stated that the grassroots typically used any informal means to reclassify rural land as industrial land as quickly as possible under such conditions. Factories were built across the PRD, and rapid urbanization occurred such that "every village has spark, and every household was smoking" [56].

Third, the informal land-use practices were driven by the economic interests of village collectives. Land rental by the grassroots increased rapidly in Dongguan in the 1990s. The village collectives borrowed loans from banks to build factories and then rented or transferred them to investors, generating significant returns for the grassroots. To encourage the expansion of the rental economy, the Dongguan City government issued *The Provisions on the Management of Rural Collective Assets in Gongguan* in 1997, and *The Measures for the Administration of House Leasing in Dongguan* in 1998. The grassroots rented their rural land or factories to investors because it generated more income than farming. In Dongguan, 70% of the village collective's income had originated from rural land or factories since the 1990s. Additionally, the grassroots earned income from "the three-processing and one compensation" factories. The proportions varied from a few to a dozen percent of the profit, and this income increased from CN 4.55¥ million in 1986 to CN 1.7¥ billion in 2002. The income was invested in building factories, constructing roads and public facilities, and funding sanitation, public security, endowment insurance, and bonuses for villagers. Rent became a stable source of income for villagers and provided maintenance funds for local public facilities.

Fourth, informal land-use practices were tolerated and given tacit approval by local governments. China began building the socialist market economic system in 1992 and reformed the tax redistribution system between the central and local government in 1994. With the decentralization of the state, those reforms increased the enthusiasm of local governments for economic growth. In this context, the government of Dongguan City projected an economic development strategy called "the second industrial revolution" in 1994 to promote rapid urban development. The Dongguan City government adopted a tolerant attitude towards informal land-use practices. According to a public statement by a spokesman of the land and resources bureau of Dongguan, the city government neither supported, encouraged, nor interfered with informal land-use practices. In fact, both the city and town level governments permitted informal land-use practices, especially the transfer of rural land prior to gaining formal approval. This led to the increased use of informal land-use practices in Changan.

In the meantime, there were two methods of informal land use adopted in Changan to cope with the re-regulation by the state. First, enterprises usually registered with the village collective and applied for land-use certificates in the name of the village collective, bypassing formal land regulations. With the rapid growth of the enterprises in Changan, the state began to monitor land use. As the transfer of rural land was strictly limited by the National Land Management Law, it became difficult for the village-township enterprises to obtain land-use rights. However, in an effort to attract more investments, the rural collectives actively helped the enterprises obtain land-use rights. A common informal practice evolved where the village collective registered the enterprise, secured the certificate of land-use rights, and then transferred the certificate to the enterprise in exchange for a "transfer fee". Because formal land institutions hindered economic development, a consensus developed among the actors in favor of this informal land-use practice.

In addition, village collectives managed rural land through the rural land stock cooperative system, which was an informal land institution innovation. The village land stock cooperative system, which evolved within the PRD in the early 1990s, successfully circumvented the National Land Management Law. On the premise of following the rule of rural land collective ownership, the rural collectives divided the collective land property rights into shares held by the villagers and initiated joint-stock companies for land

management, including the development of industrial zones, attracting FDI, and informally leasing, or transferring the rural land. Seven different modes of developing rural collective land into industrial zones are utilized in Changan, with 56 industrial zones, 1162 enterprises, and 19, 657 mu rural collectives are utilized in total in 2002 (see Table 2). However, this informal land-use practice has not yet been accepted by the state, and the land-use rights in these industrial zones have not been approved by local governments. Compared to direct land transactions between the villages and enterprises prior to the issuance of the National Land Management Law, this collective action effectively reduced or avoided the risk of rural land transfer. Economic growth in Changan was rapid under this type of informal land institution. The flexible strategy of land use in Changan successfully circumvented the state's formal land management system and served the interests of the local government, enterprises, village collectives, and villagers.

**Table 2.** The mode and number of industrial zones in Changan from 1980 to 2002.

| Development Subject and Mode | Number | Number of Enterprises | Area (mu) |
| --- | --- | --- | --- |
| Town level | 3 | 44 | 3860 |
| Village level | 25 | 372 | 8069 |
| Cooperation between village level and foreign investors | 2 | 14 | 1680 |
| Cooperation between village level and group level | 10 | 339 | 2970 |
| Cooperation between village level and private | 3 | 70 | 598 |
| Group level | 10 | 281 | 2100 |
| Cooperation between group level and private | 3 | 42 | 380 |
| Total | 56 | 1162 | 19,657 |

Source: The Annals of Changan Town.

*4.3. Institutional Innovation and the Elimination of Informal Land-Use Practices Since 2005*

Informal land-use practices have gradually become an obstacle to urban transformation and the upgrading of industries in the PRD. With globalization and the state's deregulation since the 1980s, the city-region has become the primary spatial unit participating in global competition [57]. Despite experiencing rapid economic growth after the reform and opening of China, the PRD has been faced with a series of developmental problems, such as low-end industrial structures, low land productivity, shortages of land for construction, and a large amount of informal land use. In addition, the institutional advantage the PRD once enjoyed has been lost and the region's development model has become unsustainable, placing the PRD at a disadvantage in regional and global competition. Moreover, China's central government announced its "scientific development" views and called for comprehensive, harmonious, and sustainable development in 2003, to promote city-regional transformation and productive efficiency. At the local government level, in order to return the PRD to regional competitiveness, the Guangdong Province government issued *The Outline of the Plan for the Reform and Development of the Pearl River Delta*. Along with the "dual track transformation", and the "empty the cage for new birds" development strategies in 2008, this plan was designed to foster industrial upgrades through the replacement of low-end and high-pollution manufacturing in the PRD with the addition of high-value industries.

However, formal land institutions have made it challenging to adapt to the new needs of the urban development in the PRD. Construction projects, 87.7% of which were located on rural collective land, had developed in 42.1 km$^2$ of Changan in 2005, accounting for 43% of the town's overall land. Informal industrial land, which was primarily developed without approval, accounted for 47.03% of the total industrial land area. In addition, rapid industrialization brought a large number of immigrants, many of whom rented housing from local villagers. Additionally, the development led to the emergence of informal residential land use in Changan. Nevertheless, the restrictions of the National Land Management Law on rural collective land transactions were still the primary reason for

the emergence of informal land use in Changan. The law prevented urban transformation and the upgrading of industries in the PRD, generating an urgent need for innovation in land institutions.

Land institutional innovation was led by the Guangdong Province government. In an effort to promote industrial upgrades and urban development in PRD, the government of Guangdong Province applied for central government authorization of a land institution and issued *The Measures for the Administration of Circulation of the Collective Construction Land Use Right* in 2005. The basis of this measure was the deregulation of transactions involving rural collective construction land, allowing the land to be rented and transferred at the same price as urban land. As De Soto observes, the reason for the disparity in economic development between developed and developing countries is that the former have clear formal property rights, while the latter, do not [22]. The innovation by the Guangdong Province government transformed rural collective construction land, especially some of the informal land, into assets. The rural collective construction land and part of informal land at the grassroots level in Changan was intended to be formally authorized by the state to allow this transfer of land within the formal land market. This was the first time that an informal land-use practice at the grassroots level became a foundation of institutional innovation.

Informality is the primary characteristic of the land institution modifications in the PRD. As Roy notes, urban informality is a mode of governance, and it is the production of the state within its territory [22]. The state promotes institutional innovation or issues new formal institutions by flexing its power to define and redefine formality and informality, advancing urban development, and serving political interests. Furthermore, the government of Guangdong Province announced the acceleration of regional development and industrial advancements in 2008. After the introduction of *The Measures for the Administration of Circulation of the Collective Construction Land Use Right*, the Guangdong provincial government continued to apply for central government approval of land institution innovations. It issued *Some Opinions on Implementing the 'Three Old' Transformation to Promote Economical and Intensive Land Use* in 2009, and its experimental stage lasted from 2009 to 2012. The policy stipulated that if informal land use occurred before the enforcement of the National Land Management Law commenced on January 1, 1987, villagers could apply for formal land-use rights certificates and registration, as state-owned construction land. If the informal land use occurred between January 1, 1987 and June 30, 2007, the villagers could pay a small penalty according to the National Land Management Law and then apply for a land-use rights certificate and registration as state-owned construction land. The "three old" redevelopment policy permitted villagers who desired to transfer their rural collective land to negotiate directly with developers without first reclassifying the land as state-owned construction land. The "three old" redevelopment policy availed more rural land to the market, including some informal lands that became assets for the holders. Moreover, the local city government encouraged the "three old" transformation project by returning land transfer payments to enterprises and village collectives to accelerate urban redevelopment and industrial advancement.

Led by the Guangdong provincial government, the "three old" redevelopment policy has been widely implemented in Dongguan, but as a local agent of the state, the Dongguan City government also exhibited the substantive characteristics of urban informality in the "three old" redevelopment project. For instance, one of the "three old" projects in Xiaobian village, Changan is named Ding Feng community redevelopment. The Xiaobian village collective applied for the "three old" redevelopment policy, and the city government encouraged the developers and social capital to participate in the "three old" redevelopment project during the experimental stage. The village collective directly negotiated the transfer of old informal industrial land to a developer, which formalized what had been an informal use of the rural collective land supply. Both the village collective and the developer benefitted from economic gains, and the developer had 30% of the land

transfer payment returned from the Dongguan City government for participating in the "three old" redevelopment project.

While this policy effectively promoted urban development and economic growth in Dongguan, new problems did arise. The Dongguan City government lost significant revenue due to its debasement to a passive and marginalized role. In this context, the government issued *The Operation Guidelines for the Cooperative Enforcement* of the '*Three Old' Transformation between the Collective Economic Organization and the Enterprises* in 2015, which banned direct negotiation of the transfer of rural land between village collectives and developers. Instead, land transfers were required to be conducted under the supervision of the city and township governments. The introduction of this policy meant that the rural collective land transfers between village collectives and developers became informal again, which was a reiteration of the urban informality led by the Dongguan City government as promoting land institutional change (see Table 3). The policy served the interests of the city government and would be continually adjusted on that basis.

To summarize, the spatial effect of the "three old" redevelopment policy on Changan was primarily the promotion of urban redevelopment and industrial upgrading, allowing informal land to enter the formal land market. The plan included 187 redevelopment projects covering an area of 8.29 km², and involving the industrial zones of 13 villages. Changan had completed eight "three old" redevelopment projects by 2015, covering a 0.34 km² area in which 80% of the previously industrial locations were informal land uses.

**Table 3.** The change of informal land practice in Changan since 1978.

|  | **Stage of Lack of Formal Land Institution (1978 to 1986)** | **Stage of Getting around the Formal Land Institution (1986 to 2005)** | **State of Land Institution Innovation Led by Local State (from 2005 till Now)** |
|---|---|---|---|
| Main Contradiction | Contradiction among the state and local state, the rural grassroots and land institution | Contradiction between economic development and formal land institution | Contradiction between informal land-use practice and local state |
| Agent of informal land-use practice | The state and local state and the rural grassroots | Local state and rural grassroots | Rural grassroots |
| Form of manifestation of informal land-use practice | Changing the property of land use, leasing, and transferring the land | Changing the property of land use, leasing (half-legalized), and transferring the land | Changing the property of land use (partially legalized), leasing (legalized), and transferring the land (legalized) |
| Informal land-use institution | Informal land transfer agreement (oral) | Rural land joint-stock cooperative institution | |
| Agent of institution innovation | | Villagers and rural collective | Guangdong provincial government |
| Form land institution | The Constitution in 1954 and the Household Contract Responsibility Institution | Land management law | Measures for the circulation of collective construction land and the "three old" transformation policy |

## 5. Discussion: Informality as Driver and Foundation of Institutional Innovation

*5.1. The Land Institutional Change Led by Informal Land-Use Practices*

Informal land-use practices at the grassroots level force the promulgation and enforcement of formal land institutions. At the beginning of the reform and opening, FDI drove rapid industrialization of rural areas in the PRD, and numerous acres of agricultural land were converted into non-agricultural land and used in a disordered and extensive manner. In this context, the state introduced the National Land Management Law in 1986, to regulate land use. The state allowed the transfer of the rural collective land-use rights, provided that changes to the nature of the land were approved by the local government and moved the development rights of rural collective land into the hands of the state's

agent, the governments above the county level. In addition, the state revised the constitution in 1988, separating land ownership and land-use rights, allowing the latter to be transferred in accordance with the law. While this was the first time the central government enacted market-oriented land institution reform, informal land leases and transfers had been initiated earlier at Changan in the PRD. The rapid growth of informal land led to the significant loss of agricultural lands beginning in the 1990s [6], capturing the attention of the state, which issued *the Regulations on the Protection of Basic Farmland* in 1998. This was the most rigorous farmland protection institution in the world, and it was formally implemented on 1 January, 1999. Before the enforcement of the land institution, the villagers in Changan accelerated the process of reclassifying agricultural land as non-agricultural land to capture economic benefits, leading to the loss of 6.90 km$^2$ of arable land in Changan in 1998.

### 5.2. Some Informal Land-Use Practices Were Accepted by the Local Government

Government responses to informal land-use practices in the PRD since the reform and opening have varied, in stages, between support, encouragement, toleration, acquiescence, elimination, and finally formalization. During this process, the state evolved from having no national land institution to enforcing one to allowing institutional innovation by local governments (see Figure 2). The National Land Management Law introduced in 1986 prevented any unit or individual from appropriating, selling, leasing, or transferring rural land.

The conflicts between the rigorous national formal land institution and the land-use demands of villagers in the PRD escalated in the 1990s. In response, the rural collectives divided the collective land property right into shares held by the villagers and established land stock cooperation institutions for land management, including leases of industrial lands to investors. However, land leases contained a semi-legal status under the joint-stock cooperative institution emerging in the PRD in the 1990s, which effectively circumvented the National Land Management Law, thus promoting rapid industrialization in Changan.

Because informal land-use practices made a significant contribution to the economic development in the PRD, the Guangdong Province government initiated *The Measures for the Transfer of Rural Collective Construction Land* in 2005. In order to further advance industrial upgrading and economic development in the PRD, the Guangdong Province government initiated the "three old" redevelopment policy after it was approved by the state in 2009. Informal land-use practices in the PRD were objectively accepted by both the policies, thus allowing the formalization of informal land use, which had existed for a significant period of time.

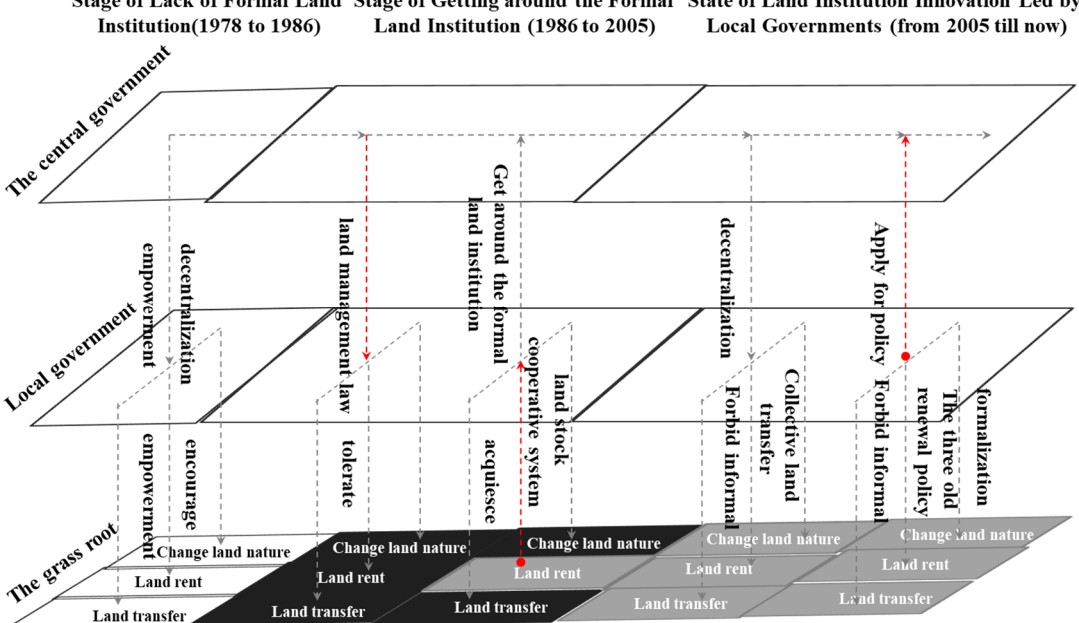

**Figure 2.** The process of formalization of informal land practice.

*5.3. The Agent of the Institution Innovation Changed from the Rural Grassroots to the Local Government at Higher Levels*

The rural land stock cooperative institution developed in the PRD in the 1990s was a typical example of the collective behavior of villagers, which was spontaneously initiated by the grassroots from the bottom up and driven by economic interests. It was an informal land institution outside the state, that ensured a steady supply of land for economic development in the PRD. On this basis, Changan developed a flourishing rental economy, thereby facilitating rapid urban development for nearly two decades, until the global financial crisis occurred in 2008 [58].

Guangdong Province began its urban development transformation and industrial upgrading strategy in 2008. However, due to strict regulations on the transfer of the rural collective land, the land supply for promoting the developmental strategy was woefully inadequate. There was a high proportion of construction land in the PRD, with the proportion of construction land in Dongguan reaching nearly 55%. On the one hand, the local government could not supply much more construction land for industrial upgrading. On the other hand, it is difficult to reclassify land that has been converted into non-agricultural land back to agricultural land. In order to address the shortage of construction land for the industrial upgrades and urban developmental transformation, the government must improve the benefit of low-output construction land and realize industrial improvements on existing informal industrial land. In this context, the Guangdong provincial government requested approval from the state for the "three old" redevelopment policy. As a land institution innovation, it facilitated the transfer of rural collective land, formalizing what had been informal land. The informal land practices and institutions that began at the grassroots level three decades ago have gradually become an institutional innovation, initially led by the Guangdong Province government and endorsed by the state.

**6. Conclusions**

This study used the case of Changan to examine the evolution of informal land-use practices. Informality in the PRD was a response to state choices in China after the reform of 1978. This paper analyzed the roots, main contradictions, and methods of the informal use practice, especially the relationship between this practice and the state, from a historical and critical governance perspective.

Informal land-use practices have experienced a range of governmental responses including encouragement, toleration, acquiescence, elimination, and finally formalization. The informality and formality of space production in Dongguan can almost be presented as a cycle that starts from an informal practice of bottom-up space production, with the empowerment of local institutions. The next step is a formal, top-down, centralized intervention, which leads to a loss of the local institutions' key role. This new situation of formality once more induces informal practices linked to the local reality, closing one cycle and probably starting another one. During this process, the state evolved from having no national land institution to enforcing one to allow institutional innovation by local governments. The land institution evolution in the PRD has been characterized by urban informality and consistent with the interests of the state. The informal land-use practices in the PRD can be regarded as the foundation of land institution innovation, which was ultimately authorized by the state.

China has gradually experienced the transformation from a planned economy to a socialist market economy since 1978. The PRD has contributed as an experimental zone for institutional innovation during this process. The state participated significantly in the process of change utilizing informal land institutions and formal institutional innovation, a classic example of reform and opening in the PRD. Roy contends that urban informality is a mode of governance and the state's production of space in its territory [24]. The state utilized its power to drive the institutional change and enact new policies to regulate the actors in urban development. Informal land-use practices are essentially the behavior of the state and its agents, which have determined the fate of the informal land-use practices through formal institutions. The new formal institutions redefine the formal and informal approaches, serving the strategies of urban development, and the needs of the state and its agents. Informal land-use practices and institutions always preceded formal land institutions in a process of continuous feedback, that promoted change in land institutions within the PRD, after the reform and opening in China.

Future studies should approach informal land-use practices and institutions rationally and objectively at the grassroots level, evaluating their impact on urban development, although some informal land-use practices have created missed opportunities. In particular, research should focus on the formalization of informality under innovation in land institutions, and the government should make a special policy for territorial lifecycle management (TLM). In Chinese governance, imperial power has not traditionally permeated the grassroots, yet, more institutional innovation is needed from the grassroots. When formal institutions are developed, earlier informal institutions may absolve, but new informality will be created. Urban development requires this sort of bottom-up institutional innovation from the grassroots.

**Author Contributions:** D.X. conceived and designed the research; Y.H. conducted this research, analyzed the data, and wrote the manuscript; G.H. revised and reformatted the overall paper. All authors have read and agreed to the published version of the manuscript.

**Funding:** This research was supported by the National Social Science Foundation of China (41901197; 41930646).

**Data Availability Statement:** The authors confirm that the data supporting the findings of this study are available within the article.

**Acknowledgments:** The authors thank the anonymous reviewers for their insightful comments and suggestions. The authors would like to thank Xiaoping Lan for their technical support.

**Conflicts of Interest:** The authors declare no conflict of interest.

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
