# Peer review of "Economic Development, Informal Land-Use Practices and Institutional Change in Dongguan, China"

_sustainability, doi:10.3390/su13042249_

Round 1

Reviewer 1 Report

The text is well structured and has a clear and logical narrative, very pleasant to read. The clarity of the speech facilitates the understanding of the studied phenomenon.

The problem addressed is well framed and presents an adequate methodology for studying the phenomenon, although it could be enriched with some quantitative data.

The authors elaborate a sound taxonomy of formality and informality in the production of urban / industrial space.

The conclusions and presentation of the case under study are liable to form the basis of a script for the interpretation of other realities of urban and industrial space production. The informality and formality of space production in Dongguan can almost be presented as a cycle that starts from an informal practice Bottom-up of space production, with the empowerment of local institutions. The next step is a formal, top-down, centralized intervention, which leads to a loss of the local institutions’ key role. This new situation of formality, once more induces informal practices linked to the local reality, closing one cycle and probably starting another one.

Reviewer 2 Report

The paper is interesting, but it should be improved, increasing the references in some parts and by better explaining the method used for the interviews. Possible improvement for authors:

Line 52: Try to add more references about the studies you "quote" without any real reference;

Lines 52-53: Try to explain which "little researches" are referred;

Line 54: the text needs references about the literature of urban informality

Lines 64-69: Try to make explicit the "existing literature";

Lines 130-131: It is necessary to quote the fundamental works by Lefebvre and Foucault about the "production of space";

Lines 254-255 and 265-267: It is necessary to explain which kind of method the authors used to interview their informer. Is it an ethnographic method? Sociological? And why?

Conclusions should be improved, discussing also the new suggeted references.

Reviewer 3 Report

Current submitted article represents extensive broad formal and informal urban territorial growth data analysis. 

Unique topic approach enables to carry on further research across different parts of China, which can feed into entire country urban informal Standard Operating Procedures (SOP) assessment.   Authors have included decent territorial regulations research and data, however latest DongGuanTerritorial Lifecycle Management (TLM) could be a beneficial subject to include within the Conclusions section.   Last but not least, some informal land-use practices have created missed opportunities. It will be a valuable aspect to consider in the future.

Round 2

Reviewer 2 Report

Dear Authors, thank you to review your article according to the suggestions. You have improved your paper and it is more coherent.